# MASIMU: Multi-Agent Speedy and Interpretable Machine Unlearning

## Abstract

The regulatory landscape around the use of personal data to train AI/ML models is rapidly evolving to protect privacy of sensitive information like user locations or medical data and improve AI trustworthiness. Practitioners must now provide the capability to "unlearn" or "forget" data—the *forget set*—that was used to train an AI model, without triggering a full model re-train on the remaining data—the *retain set* to be computationally efficient. Existing unlearning approaches train via some combination of fine-tuning pre-trained AI models solely on the retain set, pruning model weights then unlearning, and model-sparsification-assisted unlearning. In our research paper, we use deep learning (DL), multi-agent reinforcement learning (MARL) and explainable AI (XAI) methods to formulate a faster, more robust and interpretable unlearning method than past works. Our method, *multi-agent speedy and interpretable machine unlearning (MASIMU)*, fine-tunes a pre-trained model on the retain set, interpretably re-weighting the gradients of the fine-tuned loss function by computing the similarity influences of the *forget set* on the batched *retain set* based on weights generated by an XAI method. We add a MARL framework on top to address the challenge of high dimensional training spaces by having multiple agents learning to communicate positional beliefs and navigate in image environments. The per-agent observation spaces have lower dimensions, leading to the agents focusing on unlearning interpretable gradients of important superpixels that influence the target labels in the learning criteria. We provide extensive experiments on four datasets—CIFAR-10, MNIST, high resolution satellite images in RESISC-45, skin cancer images in HAM-10000 to unlearn for preserving medical privacy—computing robustness, interpretability, and speed relative to the dimensionality of the training features, and find that MASIMU outcompetes other unlearning methods.

## 1 Introduction

The large-scale adoption of Machine Learning models has led to emergence of legal opportunities where certain users would like their data to be forgotten in the training set of Artificial Intelligence (AI) models, as protected by the Right to be Forgotten (Chenou & Radu, 2019), granted by the European Union to its residents. US residents are covered by medical privacy protection under the HIPAA Federal Law (Ness et al., 2007) which is helpful to protect sensitive medical data like lung cancer images (Bandyopadhyay et al., 2021) to train AI models for cancer prediction. This helps to improve the trustworthiness of AI models. AI models will often have to follow copyright laws and regulations (Grynbaum & Mac, 2023) which can lead to the models forgetting a part of the training dataset that is subject to these copyright laws. Machine unlearning applications include lifelong learning (Liu et al., 2022), toxicity mitigation in Large Language Models (Lu et al., 2022) along with Reinforcement Learning applications (Nikishin et al., 2022; Ye et al., 2023; Guo et al., 2023).

The goal of Machine Unlearning is to effectively forget the influence of a portion of the training data, the *forget set*, on an AI model satisfying a specific objective like classification while retaining similar or better performance like the original AI model. Retraining the model from scratch on the held-out training data without including the *forget set*, called the *retain set*, takes a long time which may not be practically sustainable for AI models trained on big datasets that require high computational costs like many GPUs for training. The NeurIPS 2023 Machine Unlearning competition (Eleni Triantafillou, 2023) put forth machine unlearning evaluation criteria like unlearning taking

much less time than retraining and measuring similar performance of the unlearnt model to the original model. Another metric is success against Membership Inference Attacks (MIAs) to discern examples in the *forget set* from those in the *test set*. Existing research works perform unlearning mostly by fine-tuning pre-trained AI models on the *retain set* which poses the inherent challenge of not considering the influence of the *forget set* on the *retain set*. Latest unlearning research by pruning model weights then unlearning and with model sparsification assisted unlearning (Jia et al., 2023) improves on multi-criteria performance unlearning for a few datasets like CIFAR-10. Other unlearning related works are shared in Appendix A.1. Existing unlearning research poses significant challenges like robustness, lack of interpretability. They also do not address the unlearning problem with increasing dimensionality of training feature spaces in high-resolution images having significant amounts of information, not related to the learning objective.

We propose a baseline Machine Unlearning (MU) Framework for image classification, fine-tuning a pre-trained model on the *retain set*. For our Interpretable Machine Unlearning (IMU) Framework, we compute the *forget set* influence on the *retain set* by interpretably re-weighting the gradients of the fine-tuned loss function using similarity scores of XAI weights on the batched *retain set* and the *forget set*. XAI weights from Local Interpretable Model-Agnostic Explanations (LIME) method (Ribeiro et al., 2016), for both the *retain set* and the *forget set*, are generated faster compared to other XAI methods like SHAP scores (Lundberg & Lee, 2017) making it lucrative to be a component of our Machine Unlearning paradigm. The underlying behavior of the LIME XAI method (Garreau & Mardaoui, 2021a), like selecting local examples, identifying features and calculating weights per feature, motivate our approach for using interpretable gradients to address machine unlearning, including the use of cosine similarity and average feature weights for each label.

We formulate a Multi-agent Speedy gated recurrent unit (GRU) based Machine Unlearning (MASMU) framework with agents communicating their pose beliefs. We compare its unlearning speed with the Multi-Agent long-short term memory (LSTM) based Unlearning (MALMU). Past work, using multiple agents to classify images (Mousavi et al., 2019a), compute a spatial state positioned on each image which agents communicate to update local beliefs and policies. We combine MASMU with IMU to a Multi-Agent Speedy GRU based Interpretable Machine Unlearning (MASIMU) framework comparing it with its corresponding LSTM framework of Multi-Agent LSTM based Interpretable Machine Unlearning (MASIMU) to address the challenge of higher dimensionality for high-resolution training image features, needed to train more accurate models e.g. improving lung cancer detection (Daneshpajooh et al., 2021). The per-agent observation spaces in the MASIMU framework is small, helping to unlearn gradients of important superpixels faster that influence the probability distribution of prediction vectors in the learning criteria. We show our interpretable and robust results on the CIFAR-10, MNIST, and high resolution imagery from satellites (RESISC-45 (Cheng et al., 2017a)) and skin cancer (HAM-10000 (Tschandl et al., 2018)) data, showing improved unlearning performance with faster unlearning specially with more dimensionality on high resolution training image features using multiple agents. Our Machine Unlearning evaluation metrics (Nguyen et al., 2022) include completeness (closeness to the original model), and timeliness (time cost of unlearning as opposed to retraining). Our IMU, MASMU, MALMU, MASIMU and MALIMU unlearning frameworks are novel for high resolution image classification tasks.

## 2 RETAIN AND FORGET DATASETS

In machine unlearning, the forget dataset $D_f$ consists of a set of data items within the training dataset $D_{tr}$ for an AI model $M$ for which the influence of the data items in the *forget set* must be removed ("or unlearnt") from $M$ without full retraining on the remaining training data items, defined as the retain dataset $D_r$. Fully retraining M on $D_r$ is computationally expensive for deep learning models. A major challenge in Machine Unlearning is to learn the influence of the *forget set* on the *retain set* and to efficiently remove them from the pre-trained AI model. In our proposed MASIMU framework, we compute the influence of *forget set* and *retain set* and efficiently unlearn the influence on the pre-trained model.

We experiment on the CIFAR-10 dataset (Krizhevsky et al., 2009) with $32 \times 32$ color images having 10 labels like vehicles and animals. We also consider the MNIST dataset (Deng, 2012) with $28 \times 28$ gray-scale images of digits from 0 to 9 and their corresponding 10 labels. Finally, we unlearn AI

models on high resolution $256 \times 256$ satellite imagery in the RESISC-45 dataset (Cheng et al., 2017b) with 45 labels like mountains, houses and $450 \times 450$ skin-cancer images in the HAM-10000 dataset (Tschandl et al., 2018) with 7 labels including melanoma and melanocytic nevi, investigating unlearning of sensitive data like locations and medical records. Our train/test and the retain/forget data splits for all the datasets are provided in Table 3.

## 3 INTERPRETABLE MACHINE UNLEARNING

Feature-based XAI methods like Locally Interpretable Model-agnostic Explanations (LIME) (Ribeiro et al., 2016) are useful to explain the influence of training features on the output of an AI model, post training. LIME works by taking examples and constructing an interpretable approximate linear model around which samples are taken. For each training feature, LIME calculates $n$ weights in the following manner, where $n$ is the number of local samples used to generate the explanation as in Equation 1. LIME outputs on RESISC-45 in Figure 1 and HAM-10000 in Figure 2 show segmented superpixels on interpretable LIME masks which helps to identify similar retain and forget images.

$$w_i = e^{\frac{1}{2b^2} cosdist(\mathbf{1}, x_i)^2} \quad \forall i \le n \tag{1}$$

Here, $b$ is a bandwidth parameter, $x_i$ is a local sample selected by LIME, and $cosdist$ is the cosine distance between 2 vectors. LIME coefficients segment an image to super-pixels that can help in improving the unlearning efficiency by calculating their influence on the learning criteria.

For our baseline Machine Unlearning Framework (MU), we fine-tune the pre-trained training model on the *retain set* only, which poses the problem of not computing the influence of the *forget set* on the retain set. We define our Interpretable Machine Unlearning Framework (IMU) Algorithm weighing the influence of the LIME coefficients of the *forget set* on the batched *retain set* during fine-tuning and removing the influence of these interpretable LIME weights on the computed gradients of the super-pixels in our *retain set*. This is based on the intuition that LIME coefficient outputs are similar to the sum of the integrated gradients of the training input superpixels for AI models that are sufficient smooth in comparison to their training datasets (Garreau & Mardaoui, 2021b).

We calculate the LIME coefficient weights for each superpixel in each image and average them over each label. Then we use the $pcs$ function defined in the IMU Algorithm to compute pair-wise cosine similarity of LIME weight of every batched image in the *retain set* for batches $b$ with the LIME weights of all $f$ *forget set* images. This leads to a $b \times f$ cosine similarity matrix. We average along the rows for non-zero cosine similarity values to only compute $rsim$ the influence of *forget set* images which are more similar with *retain set* images. Then we average the similarity weightage of all the batched retain images in $crsim$ which is our interpretable weight $I_w$ highlighting the importance of the training superpixels generated by LIME. During the backpropagation of the loss function, when gradients of the loss function are computed, we update the gradients with this interpretable weight and remove their influence from the original gradients by subtraction. The computation of cosine similarity on the interpretable approximately linear LIME coefficient weights, is helpful to ensure the differentiability of the gradients of the loss function. For the unlearning problem of image classification models, we use the widely used multi-class classification loss function of Cross Entropy Loss in Equation 2. There, $x$ is a given example, $n$ is the number of classes, $y_i$ is the truth label, and $p_i(x) = \frac{e^{x_i}}{\sum_{j=1}^{n} e^{x_j}}$ is the softmax probability of $x$ being class $i$.

$$L_{CE}(x) = -\sum_{i=1}^{n} y_i \log p_i(x) \tag{2}$$

## 4 MULTI-AGENT INTERPRETABLE MACHINE UNLEARNING

A Multi-Agent (MA) Machine Unlearning (MAMU) framework has been devised in Algorithm 1 where multiple agents traverse a limited observation space based on learning the underlying belief and decision Recurrent Neural Networks (RNN) (Rumelhart et al., 1986). The RNN can be configured as either a Long Short-Term Memory (LSTM) (Hochreiter & Schmidhuber, 1997) or a Gated Recurrent Unit (GRU) (Chung et al., 2014) in the MALMU (Multi-Agent LSTM based Machine Unlearning) framework or the MASMU (Multi-Agent Speedy GRU based Machine Unlearning)

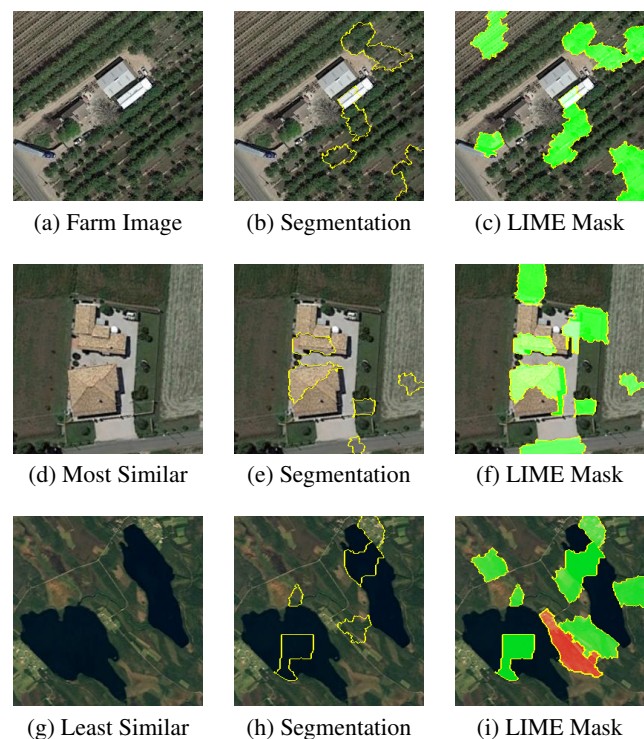

(a) Farm Image    (b) Segmentation    (c) LIME Mask

(d) Most Similar    (e) Segmentation    (f) LIME Mask

(g) Least Similar    (h) Segmentation    (i) LIME Mask

Figure 1: (a) Farm Image from RESISC-45 *retain set* (d) Farm Image from RESISC-45 *forget set* with its LIME coefficient vector and that of the original image having the highest cosine similarity (g) Lakes Image from RESISC-45 *forget set* with its LIME coefficient vector and that of the original image having the lowest cosine similarity.

framework. We use the RNN to represent belief that is propagated across the agents per step with the incentive of speeding up image unlearning using MA-REINFORCE algorithm. For unlearning, we load the model trained with MA-REINFORCE algorithm on the entire dataset and fine-tune it using the *retain set* with MA-REINFORCE to unlearn the *forget set* images. This MA framework can improve unlearning for high resolution images, e.g. RESISC-45 satellite images, reducing the dimensionality with less observation dimensions per agent. MALMU and MASMU are inspired by the training of MARL algorithms (Mousavi et al., 2019b) using LSTM RNNs classifying high resolution images.

We model our MAMU frameworks, MALMU and MASMU, as Partially Observable Markov Decision Processes (POMDPs). For an agent classifying an image $I$, the state consists of the position of the center of the agent on $I$, as well as the history of the belief RNN and the decision RNN. When GRUs are used to compute the beliefs, only a hidden state is updated. Actions available to the agent are to move the position a pixel up, down, left or right. This is constrained by the requirement that the agent's observation window fits entirely within $I$. Using a window size reduces the dimensionality of the observation space when supplied with high-resolution images like in RESISC-45 or HAM-10000. Transitions come from policy function $\pi$ conditioned on the hidden cell state of the decision RNN in the case of updates to the spatial state (position), and from the output of a parameterized function supplied with the previous state and information input in the case of the belief and decision RNNs. Differentiable rewards across classification model network parameters are calculated by taking the difference of a random loss and cross entropy loss at each step. A detailed mathematical discussion on computing LSTM and GRU based belief RNNs along with corresponding decision RNNs to sample actions based on the policy gradients of MA-REINFORCE algorithm has been shared in the Appendix A.4.

For an image $I$ with ground truth label $i \in \{1 \ldots M\}$, to incentive speedy unlearning, rewards for a particular trajectory with positive probability $\tau$ are calculated by grouping the various parameters in our algorithm into one single parameter $\Theta$. The differentiable reward across network parameters

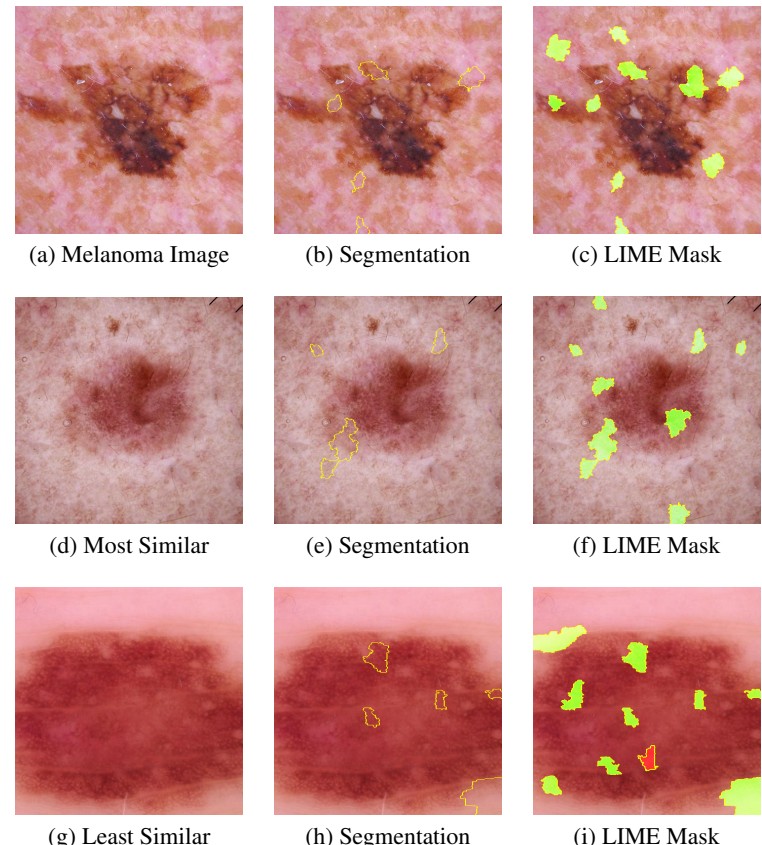

(a) Melanoma Image     (b) Segmentation     (c) LIME Mask

(d) Most Similar     (e) Segmentation     (f) LIME Mask

(g) Least Similar     (h) Segmentation     (i) LIME Mask

Figure 2: (a) (Melanoma Image from HAM-10000 *retain set* (d) A Melanoma Image from HAM-10000 *forget set* for which its LIME coefficient vector and that of the original image has the highest cosine similarity (g) Melanocytic Nevi Image from HAM-10000 *forget set* for which its LIME coefficient vector and that of the original image has the lowest cosine similarity.

$r_\tau$ is computed in Equation 3 to follow the aggregate prediction of the agents where $e_i$ is the unit vector along the ground truth's direction.

$$r_\tau = -L(\bar{p} - e_i) \tag{3}$$

Our multi-agent learning paradigms use a LSTM RNN in MALMU similar to the same MA-REINFORCE approach as (Mousavi et al., 2019a) with also while using a GRU RNN in MASMU. For MALIMU and MASIMU, we update the parametric gradients of the loss function, used to compute the differentiable rewards for MA-REINFORCE algorithm, by subtracting, thereby removing the interpretable weight of similar super-pixels in the retain set and the forget set which is computed using LIME interpretable AI method as shown in Algorithm 2. Our goal is to adjust the parameters of our system $\Theta$ in such a way that we maximize the objective function in Equation 4.

$$J(\Theta) = \mathbb{E}_{\tau \in \mathrm{T}}[r_\tau] \tag{4}$$

T is the set of possible trajectories for our agents. The original REINFORCE algorithm (Sutton et al., 2000), extended to MA-REINFORCE, computes gradients of the objective function in Equation 5.

$$\nabla J(\Theta) = \mathbb{E}[\sum_{\tau \in \mathrm{T}} \nabla(\log p_\tau)r_\tau + \nabla r_\tau] \tag{5}$$

which can be approximated with an unbiased estimator for $J$, obtained by sampling $N$ trajectories:

$$\hat{J}(\Theta) = \frac{1}{N}\sum_{i=1}^{N}(\log p_{\tau_i})r_{\tau_i}^d + r_{\tau_i} \implies \mathbb{E}[\nabla\hat{J}(\Theta)] = \nabla J(\Theta) \tag{6}$$

---

**Algorithm 1** Multi-Agent Machine Unlearning (MAMU) (for both MALMU and MASMU)

---

1: **Input:** retain data $D_r$ of size $r$, pre-trained model $M$
2: **Training Parameters:** epochs $e$, batches on retain data $b$, loss function $L_f$, optimizer $O$, batch size $g$, agents $N$, steps $T$
3: Initialize $M_u = M$
4: **for** $i = 1$ **to** $e$ **do**
5:    **for** $j = 1$ **to** $b$ **do**
6:       **for** $k = 1$ **to** $g$ **do**
7:          **for** $v = 1$ **to** $N$ **do**
8:             Initialize $s_v(0)$ on a random pixel in image $I_k$
9:             Initialize $h_v(0) = 0, c_v(0) = 0$
10:            **for** $w = 1$ **to** $|\mathcal{N}_v|$ **do**
11:               $m_w(0) = 0$
12:            **end for**
13:          **end for**
14:          **for** $t = 0$ **to** $T - 1$ **do**
15:            **for** $v = 1$ **to** $N$ **do**
16:               Make observation $o_v(t) = \text{observe}(I_k, s_v(t))$
17:               Get feature extraction $b_v(t) = \mathbf{b}_{\theta_4}(o_v(t))$
18:               Get state representation $q_v(t) = \mathbf{q}_{\theta_5}(s_v(t))$
19:               Calculate aggregate message $\bar{d}_v(t) = \frac{1}{\text{in-deg}(v)} \sum_{n=1}^{N} d_n(t)$
20:               Form information input $u_v(t) = [b_v(t)^T \ q_v(t)^T \ \bar{d}_v(t)^T]$
21:               Run belief RNN using $u_v(t)$ as input
22:               Generate message $m_v(t) = \mathbf{m}_{\theta_2}(h_v(t))$
23:               Run decision RNN using $u_v(t)$ as input
24:               Update policy $\pi$ on $\pi_{\theta_5}(\cdot, \hat{h}_v(t+1))$
25:               Get action $a_v(t+1)$ from $\pi$
26:               Go to new spatial state $s_v(t+1) = \text{transition}(s_v(t), a_v(t+1))$
27:            **end for**
28:          **end for**
29:          **for** $v = 1$ **to** $N$ **do**
30:            Generate prediction vector $p_v$
31:          **end for**
32:          Calculate mean prediction vector $\bar{p}$
33:          Compute discounted differentiable rewards with MA-REINFORCE policy gradients in Equation (6) and update parameters
34:       **end for**
35:    **end for**
36: **end for**
37: **return** $M_u$

---

$r_{\tau_i}^d$ in Equation 6 denotes the reward of sampled trajectory $\tau_i$ detached from the computational graph and treated as a scalar, as in (Mousavi et al., 2019a).

The resilient performance with increasing dimensionality of high resolution training images, allows our MASMU framework to be applicable in scenarios requiring the unlearning of very detailed and therefore potentially extremely sensitive images. This also motivates our framework of Multi-Agent Speedy and Interpretable Machine Unlearning (MASIMU) as described in Algorithm 2, interpretably unlearning the influence of specific super-pixels in high resolution sensitive images by re-weighting the gradient weights during fine-tuning just like in the IMU Algorithm without multiple agents. After the agents are initialized, MASIMU algorithm goes through a number of steps where information is exchanged between agents through the use of a RNN structure like LSTM or GRU. Using this information as well as an observation of the immediate environment (pixels of the image in the neighborhood of the agent), the agent makes a prediction of an image's class and then takes an action. After each batch of predictions, losses are calculated for the policy-based actor deciding which action to take and the critic network assigning values to the actions taken by each agent, and their weights are updated accordingly.

---

**Algorithm 2** Multi-Agent Interpretable Unlearning (MAIMU) (for both MALIMU and MASIMU)

---

1: **Input:** training data $D_{tr}$ of size $tr$, test data $D_{te}$ of size $te$, retain data $D_r$ of size $r$, forget data $D_f$ of size $f$, LIME coefficients on retain data $I_{D_r}$, LIME coefficients on forget data $I_{D_f}$, baseline model $M$
2: **Note:** $f = tr - r$
3: **Training Parameters:** epochs $e$, batches on retain data $b$, loss function $L_f$, batch size $g$, agents $N$, steps $T$
4: $M_u = M$
5: **repeat**
6:     **for** $i = 1$ **to** $e$ **do**
7:         **repeat**
8:           **for** $i = 1$ **to** $b$ **do**
9:             Use $N$ agents to run an episode of MA-REINFORCE as in Algorithm 1
10:             Obtains batch input features $D_r^{bf}$ and target labels $D_r^{bt}$ for $b_r$ batched images
11:             Obtains LIME scores for batched images $I_{D_r^b}$
12:             Note: LIME scores are $\sum$ of interpretable gradients over batched superpixels
13:             Clears gradients of parameters in $M$ tracked by $O$
14:             $sim = \text{pcs}(I_{D_r^b}, I_{D_f})$
15:             $rsim = \text{rowwise\_average}(sim \text{ for } sim \neq 0)$
16:             $crsim = \text{columnwise\_average}(rsim)$
17:             $I_w = crsim$ (Interpretable weight of similar super-pixels in retain & forget sets)
18:             $output = M_u(D_r^{bf})$
19:             $loss = L_f(output, D_r^{bt})$
20:             Compute gradients $\nabla(loss)$ on the loss function during backward propagation
21:             Note: There are $p$ parameters in the pre-trained model
22:             Note: Interpretably unlearning influence of retain set on the forget set
23:             **repeat**
24:                 **for** $i = 1$ **to** $p$ **do**
25:                   $\nabla_p(loss) = \nabla_p(loss) - I_w * \nabla_p(loss)$
26:                 **end for**
27:             **until** all $\nabla_p(loss)$ are updated
28:           **end for**
29:         **until** all $b$ batches are processed
30:     **end for**
31: **until** all $e$ epochs are updated
32: Returns $M_u$ unlearnt model

---

We measure the unlearning accuracy and loss (as in Equation 2) on AI models classifying each dataset to measure the quality and accuracy of our unlearning frameworks. For successful unlearning, it is good for the accuracy of the unlearned model on the *forget set* to be close to the accuracy of the unlearned model on the test set, as it indicates that the "forgotten" examples have never been seen by the model to begin with, just like the test samples. Similarly, unlearned model on the *retain set* should have a similar accuracy to the training accuracy of the original model. We also measure completeness (Cao & Yang, 2015) of how close the unlearned model is to the original model. A high completeness indicates better unlearning where the unlearned model is less distinguishable from the original model when evaluated on unseen examples. We use the accuracy of Membership Inference Attacks (MIAs) (Shokri et al., 2017), to measure how successfully we can guess that a given example is part of *retain set* or *forget set*. Details of the above metrics are described in Appendix A.5.

## 5 RESULTS

We train our Machine Unlearning (MU), Interpretable Machine Unlearning (IMU), Multi-Agent Speedy MU (MASMU) and Multi-Agent Speedy IMU (MASIMU) frameworks using a stochastic gradient descent (SGD) optimizer, a learning rate of 0.1 and a cross-entropy loss function. A comparative analysis of the unlearning performance for IMU and MU frameworks on the low-dimensional

Table 1: Unlearning Results for Baseline Unlearning (MU) and our Interpretable Unlearning IMU Experiments (Exp) on Completeness (Comp) and Membership-Inference Attack (MIA) metrics

| EXPERIMENT | DATASET | MIA | COMP |
|---|---|---|---|
| MU | RESISC-45 | 0.532 | 0.817 |
| IMU | RESISC-45 | 0.536 | 0.813 |
| MU | HAM-10000 | 0.640 | 0.781 |
| IMU | HAM-10000 | 0.640 | 0.760 |
| MU | CIFAR-10 | 0.503 | 0.832 |
| IMU | CIFAR-10 | 0.513 | 0.849 |
| MU | MNIST | 0.545 | 0.998 |
| IMU | MNIST | 0.545 | 0.995 |

images in MNIST and the high-dimensional satellite images in RESISC-45 datasets indicates improved unlearning with increasing accuracy and decreasing loss across 25 epochs for IMU in Figure 3 when the gradients are interpretably re-weighted. Table 1 indicates that IMU framework is better for unlearning, increasing the completeness measure and taking the MIA accuracy score closer to 0.5 in comparison to MU. MASIMU computes local beliefs with GRU. Local beliefs are also computed with LSTMs in MALIMU framework for comparative analysis. Figure 5 shows that the benefits of using Multiple agents on unlearning time scale with the dimensionality of the dataset. MASIMU and MALIMU outperform all other frameworks on the very high dimensional HAM-10000 dataset and are not far behind in the high dimensional RESISC-45 dataset. Comparison of with retraining on retain set from scratch can be found in Table 4 showing that retraining from scratch is slower.

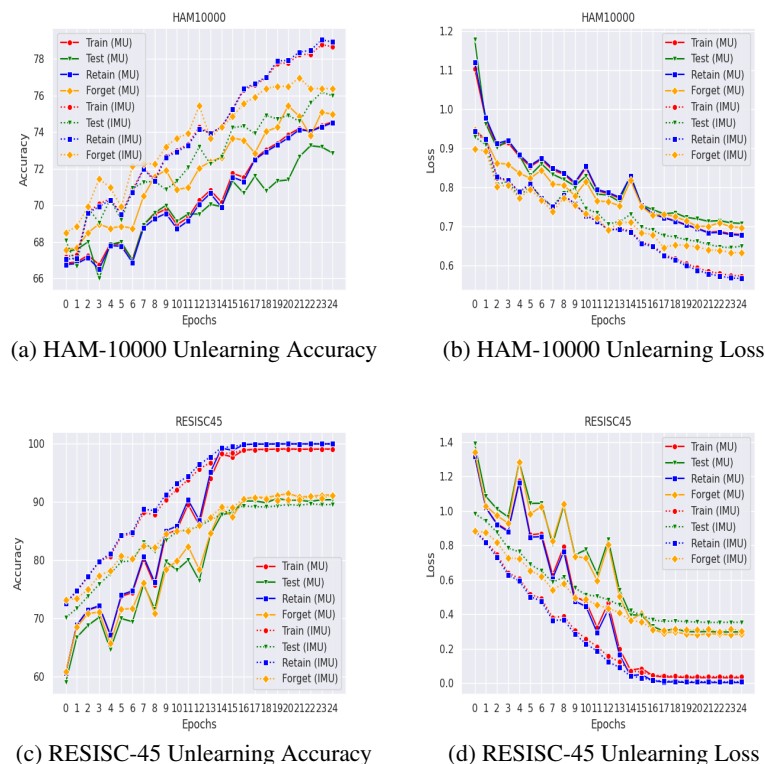

(a) HAM-10000 Unlearning Accuracy

(b) HAM-10000 Unlearning Loss

(c) RESISC-45 Unlearning Accuracy

(d) RESISC-45 Unlearning Loss

Figure 3: Machine Unlearning (MU) and Interpretable MU (IMU) Accuracy and Loss Plots

For our MASMU & MASIMU experiments, we use 3 agents, 5 steps per episode, an observation window size of 6, and a learning rate of $1 \cdot 10^{-3}$, for the low-resolution MNIST images. On higher resolution images like RESISC-45, we use 16 agents, 16 steps per episode, an observation window size of 12, and a learning rate of $1 \cdot 10^{-4}$. We reduce the learning rate for our multi-agent frameworks to make smaller learning steps by multiple agents for the optimal solution. This leads to the MASIMU and MASMU comparison over 5 epochs in Figure 4 on MNIST and RESISC-45 datasets

Table 2: Unlearning Results for our Baseline Multi-Agent Speedy Machine Unlearning (MASMU) and our Multi-Agent Speedy and Interpretable Machine Unlearning Framework (MASIMU) on Completeness (Comp), and Membership-Inference Attack (MIA) metrics.

| EXPERIMENT | DATASET | BELIEF | MIA | COMP |
|---|---|---|---|---|
| MALMU | RESISC-45 | LSTM | 0.531 | 0.615 |
| MASMU | RESISC-45 | GRU | 0.531 | 0.596 |
| MALIMU | RESISC-45 | LSTM | 0.538 | 0.595 |
| MASIMU | RESISC-45 | GRU | 0.533 | 0.603 |
| MALMU | HAM-10000 | LSTM | 0.640 | 0.828 |
| MASMU | HAM-10000 | GRU | 0.640 | 0.807 |
| MALIMU | HAM-10000 | LSTM | 0.640 | 0.814 |
| MASIMU | HAM-10000 | GRU | 0.640 | 0.838 |
| MALMU | CIFAR-10 | LSTM | 0.498 | 0.645 |
| MASMU | CIFAR-10 | GRU | 0.501 | 0.647 |
| MALIMU | CIFAR-10 | LSTM | 0.498 | 0.648 |
| MASIMU | CIFAR-10 | GRU | 0.501 | 0.636 |
| MALMU | MNIST | LSTM | 0.545 | 0.756 |
| MASMU | MNIST | GRU | 0.545 | 0.729 |
| MALIMU | MNIST | LSTM | 0.545 | 0.770 |
| MASIMU | MNIST | GRU | 0.545 | 0.732 |

showing a trend of MASIMU being better at unlearning in comparison to MASMU. Completeness increases and MIA values are closer to 0.5 in case of MASIMU as shown in Table 2.

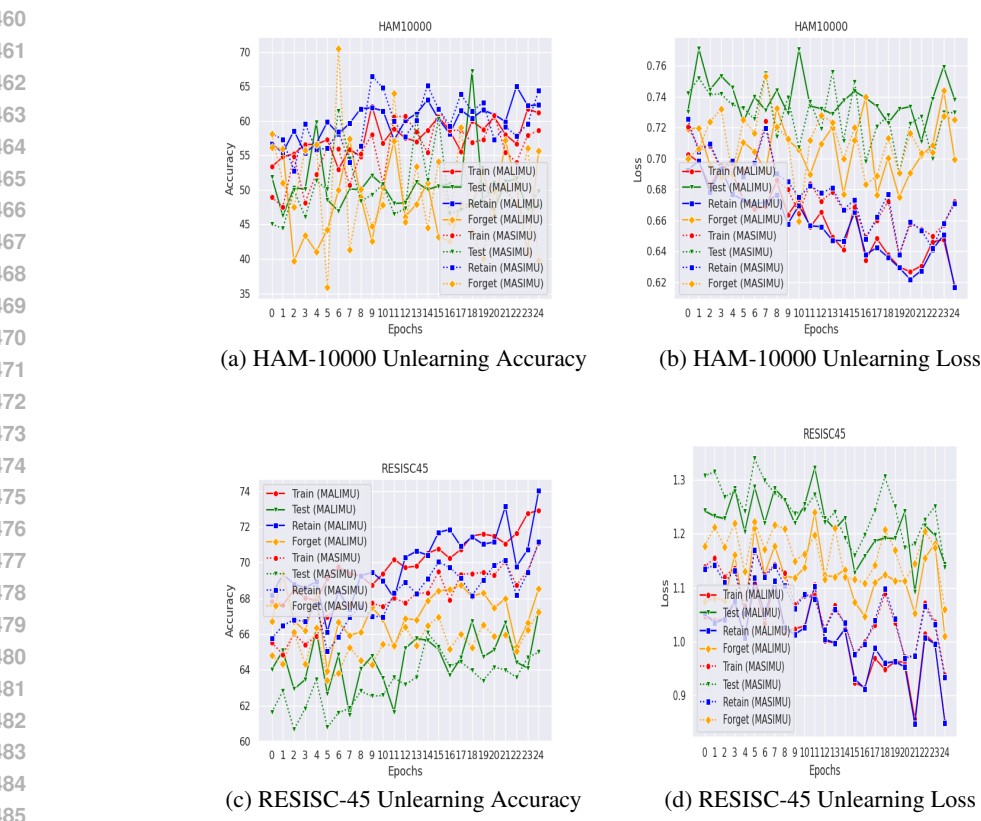

(a) HAM-10000 Unlearning Accuracy   (b) HAM-10000 Unlearning Loss

(c) RESISC-45 Unlearning Accuracy   (d) RESISC-45 Unlearning Loss

Figure 4: Multi-Agent Unlearning Accuracy and Loss Plots

The accuracy on the *forget set* is comparable with that on the *test set* for the interpretative unlearning frameworks, showing robustness for IMU, MALIMU or MASIMU. This achieves a major part of the unlearning objective, which is that a member of the *forget set* should be evaluated as if the model had never seen it in the first place. Unlearning time significantly reduces for MASMU, MALMU, MASIMU and MALIMU, on the HAM-10000 and RESISC-45 datasets with respect to IMU and MU as shown in Figure 5. For HAM-10000, GRU-based MASMU is faster than LSTM-based MALMU while MASIMU is slightly faster than MALIMU. For RESISC-45, MASMU and MALMU have comparable unlearning times and so do MALIMU and MASIMU with MALMU being slightly faster than MASMU. More importantly, unlearning time significantly decreases for MALIMU and MASIMU in comparison to MALMU and MASU, even with the additional computational cost of interpretatively re-weighting gradients during back-propagation in the fine-tuning process. Multiple agents reduce observation space dimensionality per agent, leading to faster unlearning which is important for AI applications sensitive to latency like disaster management detecting satellite images (e.g. RESISC-45) or protecting medical privacy in skin cancer images (e.g. HAM-10000).

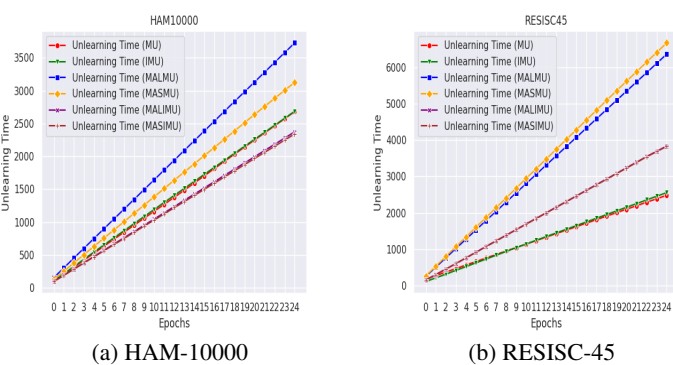

(a) HAM-10000        (b) RESISC-45

Figure 5: Unlearning Time Plots comparing our Frameworks

## 6 CONCLUSION AND FUTURE WORK

We have presented a Machine Unlearning (MU) framework with an interpretative component (IMU) that we have extended with multiple agents (MA) in MALMU, MASMU, MALIMU and MASIMU frameworks. Interpretation is important for providing insights into the behavior of the unlearned model so that we understand how our model is unlearning by forgetting the influence of major superpixels of images in the *forget set*, a part of the original training set. Our results show that Interpretable Machine Unlearning (IMU) is better than fine-tuning on the *retain set* (MU) when it comes to completeness and accuracy on an MIA. When it comes to increased dimensionality with high resolution training examples, MALIMU, MASIMU, MALMU and MASMU frameworks are significantly faster than IMU and MU for the MNIST and RESISC-45 datasets, which is an important factor in weighing the compute costs and benefits of unlearning versus retraining a model. Notably MASIMU and MALIMU are both faster than MALMU and MASMU, even with added compute cost for interpretability, showing that multiple agents unlearn faster by reducing observation space per agent. Furthermore, both IMU, MALIMU and MASIMU share the desirable robustness property that an unlearned model has similar accuracy on the *forget set* and the test set, leading to a decreased probability that an adversary can use performance of the model on a member of the *forget set* to infer membership in the training set. In future, we hope to explore how state-of-the-art cooperative decision making algorithms such as Proximal Policy Optimization (PPO) (Schulman et al., 2017) and Multi-Agent PPO (MAPPO) (Yu et al., 2022) along with single agent algorithms like self-play (Bai et al., 2020) can be used to further increase the unlearning performance of MASIMU.

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

# A APPENDIX

## A.1 ADDITIONAL RELATED WORKS

Machine Unlearning is a formulated as a problem (Nguyen et al., 2022) where there is a dataset $D$, a *forget set* $D_f \subset D$, and a model trained on the dataset $A(D)$ passed into an unlearning algorithm $U(\cdot)$. The unlearning algorithm returns a model where the influence of the members of $D_f$ on the output of the model has been reduced. Reasons are given to motivate the task of machine unlearning, namely the removal of sensitive data from models used in sensitive industries such as healthcare and finance. Several challenges arise when tackling machine unlearning, such as the stochasticity of many training methods and reduction in performance for models that have been unlearned. The survey also posits some desired properties of unlearning algorithms: having similar accuracy to the original model (completeness) and being fast enough to justify not retraining the model (timeliness). These two properties have a trade-off that must be considered when deciding whether to retrain or unlearn a model. A summary and comparison of many unlearning methods is provided, covering different types of methods (model-agnostic, model-intrinsic, data-driven), scenarios where the methods can be applied (few-shot, zero-shot, zero-glance, exact, approximate), properties of the methods (completeness, timeliness, etc.) and the kinds of data that can be unlearned (items, features, etc.). No reinforcement-learning based or multi-agent unlearning method was mentioned in this survey, unlike our novel multi-agent reinforcement learning frameworks.

Machine unlearning has been applied to a wide variety of settings, including lifelong learning (Liu et al., 2022) and toxicity mitigation in Large Language Models (Lu et al., 2022). This includes applying it to Reinforcement Learning (Nikishin et al., 2022) (Ye et al., 2023) (Guo et al., 2023). Unlike prior work, however, we are focused on using Reinforcement Learning to forget examples from a subset of a training set rather than having agents unlearn deleterious behavior learned early on in training, attempting to forget an environment, or mitigating attacks by a trojan agent. Multi-objective Reinforcement Learning has been discussed as a possible future direction for machine unlearning (Kassem et al., 2023) but has not yet been attempted as far as we are aware.

(Laroche & Tachet Des Combes, 2022) address the issue of unlearning bad convergences when making policy updates in Reinforcement Learning. It does not really have anything to do with machine unlearning as the problem is formulated in works like (Nguyen et al., 2022). Nonetheless, it proposes to speed up the unlearning process through a modified cross-entropy-based approach, in contrast to traditional policy gradient updates.

In order to address the problem of models that deal with outdated, irrelevant, or private data, (Shaik et al., 2023) introduce FRAMU, a framework that uses Reinforcement Learning and Federated Learning to achieve machine unlearning. Attention-based Machine Unlearning using Federated Reinforcement Learning. FRAMU can work with both single-modal and multi-modal data, and is suited for situations where the data distribution is dynamic. However, FRAMU is not very scalable and is computationally complex.

There have been attempts to apply machine unlearning to multi-modal data with potentially dynamic data distributions (Shaik et al., 2023) but so far they have not been scalable or computationally complex. As we value time complexity, we only consider single-modal data.

An approach to machine unlearning by sparsifying model parameters is posited by (Jia et al., 2023). While this research work achieves good results in metrics such as accuracy and Membership Inference attacks using resnet18 models and the CIFAR-10 dataset, the time cost associated with making the models sparse makes it less appealing as an unlearning method, given the possibility of retraining the model from scratch if time is not a concern. Furthermore, unlearning through making models sparse is not interpretable. These unlearning methods are not applicable for high resolution image classification tasks.

## A.2 STATISTICS OF RETAIN AND FORGET SETS

Table 3: Statistics on the train, retain, forget and test datasets

| DATASET | TRAIN | RETAIN | FORGET | TEST |
|---------|-------|--------|--------|------|
| RESISC-45 | 26775 | 24098 | 2677 | 4725 |
| HAM-10000 | 8512 | 7661 | 851 | 1503 |
| CIFAR-10 | 50000 | 45000 | 5000 | 10000 |
| MNIST | 60000 | 54000 | 6000 | 10000 |

## A.3 ALGORITHM FOR INTERPRETABLE MACHINE UNLEARNING

## A.4 COMMUNICATION IN MULTI-AGENT MACHINE UNLEARNING

For MASMU, our agents are represented via vertices in a directed graph $\mathcal{G}$ for $N$ agents denoted by $\{1, ..., N\}$, the state of agent $i \in N$ at step $t$ by $s_i(t)$, the observation of agent $i$ at step $t$ by $o_i(t)$, and the sampled action of agent $i$ at time step $t$ by $a_i(t)$. The set of edges in $G$ is given by $\mathcal{E} \subset \{(i, j) : i \neq j\}$, where $(i, j) \in \mathcal{E}$ represents that $i$ communicates messages to $j$. We let $\mathcal{N}_i$ denote the set of neighbors of $i$, i.e., $\mathcal{N}_i = \{j : (i, j) \in \mathcal{E}\}$. An RNN is used to calculate the belief of the agent as it progresses through the task. We denote the hidden state of agent $i$'s belief LSTM at time step $t$ via $h_i(t)$, and similarly denote the cell state (if the RNN is an LSTM) with $c_i(t)$. The hidden state of the belief RNN is used to create a message in Equation 7

$$m_i(t) = \mathbf{m}_{\theta_1}(h_i(t)) \tag{7}$$

where $\mathbf{m}_{\theta_1}$ is a function parameterized on $\theta_1$. This message is shared with agents in $\mathcal{N}_i$. An agent receives its messages from its neighbors and decodes them via a trainable parameterized function

$$d_i(t) = \mathbf{d}_{\theta_2}(m_i(t)) \tag{8}$$

with $\theta_2$ parameters on d, aggregated by averaging to get

$$\bar{d}_i(t) = \frac{1}{indeg(i)} \sum_{j=1}^{N} d_j(t) \tag{9}$$

where $indeg(i)$ is the number of nodes in $G$ pointing to $i$. Features are extracted from the local observation by a trainable function

$$b_i(t) = \mathbf{b}_{\theta_3}(o_i(t)) \tag{10}$$

where $\theta_3$ represents the parameters of $\mathbf{b}$. We prepare the position of $i$ for input to the belief RNN through parameterized mapping and thereby update the belief RNN.

$$q(t) = \mathbf{q}_{\theta_4(s_i(t)}. \tag{11}$$

If the RNN is an LSTM, it is updated according to the Equation 12.

$$\begin{bmatrix} h_i(t+1) \\ c_i(t+1) \end{bmatrix} = \mathbf{f}_{\theta_5}(\begin{bmatrix} h_i(t) \\ c_i(t) \end{bmatrix}, u_i(t)) \tag{12}$$

---

**Algorithm 3** Interpretable Machine Unlearning (IMU)

---

1: **Input:** training data $D_{tr}$, test data $D_{te}$, retain data $D_r \subseteq D_{tr}$, forget data $D_f = D_{tr} \setminus D_r$, LIME coefficients on retain data $I_{D_r}$, LIME coefficients on forget data $I_{D_f}$, baseline model $M$.

2: **Training Parameters:** epochs $e$, batches on retain data $b$, loss function $L_f$, learning rate scheduler $LS$, optimizer $O$

3: $M_u = M$

4: **repeat**

5:    **for** $i = 1$ **to** $e$ **do**

6:       **repeat**

7:          **for** $i = 1$ **to** $b$ **do**

8:             Obtains batch input features $D_r^{bf}$ and target labels $D_r^{bt}$ for $b_r$ batched images

9:             Obtains LIME scores for batched images $I_{D_r^b}$

10:           Note: LIME scores are $\sum$ of interpretable gradients over batched superpixels

11:           Clears gradients of parameters in $M$ tracked by $O$

12:           $sim = \text{pcs}(I_{D_r^b}, I_{D_f})$

13:           $rsim = \text{rowwise\_average}(sim \text{ for } sim \neq 0)$

14:           $crsim = \text{columnwise\_average}(rsim)$

15:           $I_w = crsim$ (Interpretable weight of similar super-pixels in retain & forget sets)

16:           $output = M_u(D_r^{bf})$

17:           $loss = L_f(output, D_r^{bt})$

18:           Computes gradients $\nabla(loss)$ on the loss function during backward propagation

19:           Note: There are $p$ parameters in the pre-trained model

20:           Note: Interpretably unlearning influence of retain set on the forget set

21:           **repeat**

22:             **for** $i = 1$ **to** $p$ **do**

23:                $\nabla_p(loss) = \nabla_p(loss)$ - $I_w * \nabla_p(loss)$

24:             **end for**

25:           **until** all $\nabla_p(loss)$ are updated

26:         **end for**

27:       **until** all $b$ batches are processed

28:    **end for**

29: **until** all $e$ epochs are updated

30: Returns $M_u$ unlearnt model

---

.

If the RNN is a GRU then the update question will take the form:

$$h_i(t + 1) = \mathbf{f}_{\theta_5}(h_i(t), u_i(t)) \tag{13}$$

.

where $\mathbf{f}_{\theta_5}$ is a trainable function, $u_i(t) = [b_i(t)^T \ \bar{d}_i(t)^T \ q_i(t)^T]$ consists of a three-part information input containing extracted features from the local observation, a representation of the agent's position within the example image, and the aggregate of the messages received by $i$. A decision LSTM with hidden state $\hat{h}_i(t)$ and cell state $\hat{c_i}(t)$ is used for updating the policy.

$$\pi(a) = \pi_{\theta_6}(a, \hat{h}_i(t + 1)) \tag{14}$$

for action $a \in \mathcal{A}$. The decision LSTM is updated using the same information as the belief LSTM in Equation 15.

$$\begin{bmatrix} \hat{h}_i(t+1) \\ \hat{c}_i(t+1) \end{bmatrix} = \mathbf{f}_{\theta_7}(\begin{bmatrix} h_i(t) \\ \hat{c}_i(t) \end{bmatrix}, u_i(t)) \tag{15}$$

If a decision GRU is used instead of an LSTM, we only need to update the hidden state with Equation 16.

$$\hat{h}_i(t + 1) = \mathbf{f}_{\theta_7}(h_i(t)u_i(t)) \tag{16}$$

We can sample an action $a_i(t)$ from the action space using $\pi$, and update our spatial state $s_i(t + 1)$ accordingly. Each agent generates a raw prediction vector per step, with a value for each class using

parameterized mapping $p_i = \mathbf{p}_{\theta_8}(c_i(T))$. Finally, we calculate the shared prediction vector by averaging the raw prediction vectors across the agents. Thus our agents collaborate to form a final prediction vector.

### A.5 MULTI-AGENT MACHINE UNLEARNING EVALUATION

In order to evaluate the Multi-Agent Machine Unlearning frameworks, we compute the accuracy of the AI models, on the *retain set* $D_{ret}$, the *forget set* $D_f$, the *test set* $D_{te}$ and the training set $D_{tr}$. For a given dataset $D$ and AI model $M$, we calculate accuracy by calculating the number of correct predictions by $M$ on $D$ and then dividing it by the number of examples in $D$. Multi-label classification losses on each dataset are calculated using standard Cross Entropy Loss in Equation 2.

Another important comparison to make is the predictions of the unlearned model itself with those of the original model using distance metrics to quantify how "close" these two models are. To assess how often the predictions made by the original model $M$ align with the predictions made by an unlearned model $U$, we calculate the "completeness" of the unlearned model in Equation 17. There, $X$ is the *test set* of examples and $\mathbf{1}_{\{C(U(x))\}}C(M(x))$ is the indicator function equal to one when the predicted class of the unlearned model $U$ is the same as that of the original model $M$, and zero otherwise.

$$completeness(U, M) = \frac{\sum_{x \in X} \mathbf{1}_{\{C(U(x))\}}C(M(x))}{|X|} \qquad (17)$$

The concept of computing the accuracy of Membership Inference Attacks (MIA) (Shokri et al., 2017) lends itself naturally to unlearning, where given a model $M$ and a combination of examples from the *forget set* $D_f$, and *test set* $D_{te}$, we see how accurately we can distinguish examples used to train the model (members of the forget set) and examples not seen by the model during training (the test set). Intuitively, we want the MIA's accuracy closer to $0.5$ (a random guess). Our baseline models fine-tuned on the *retain set* $D_r$ are scored in Equation 18 with a simple MIA attack consisting of 10-fold cross-validation score for a simple logistic regression model trained on losses from samples of $D_f$ and $D_{te}$ and categorized based on their inclusion in the training set (i.e. members of the *forget set* are also in the training set, whereas member of the *test set* are not).

$$MIA(SL, inTrain) = CV_{10}(LR, SL, inTrain) \qquad (18)$$

where $SL \in \mathbb{R}^n$ is a set of losses for $n$ examples, $inTrain \in \{0, 1\}^n$ is a set of 0s and 1s categorizing whether the example is in the training set (1) or not (0), $LR$ is a logistic regression model, and $CV_{10}$ is a standard 10-fold cross validation procedure that returns an accuracy score of how well the logistic regression model distinguishes the members of the *forget set* from the *test set* based on their example losses.

### A.6 RETRAINING TIME ON RETAIN SET

| Dataset | Multi-Agent Retraining Time (seconds) |
|---------|---------------------------------------|
| RESISC-45 | 5291.96 |
| HAM10000 | 3556.20 |

Table 4: Multi-Agent Retraining Time on Retain Set re-training from scratch (25 Epochs)

