# OpenReview forum: "MASIMU: Multi-Agent Speedy and Interpretable Machine Unlearning"
_ICLR.cc/2025/Conference — ICLR 2025 Conference Withdrawn Submission_

### Official Review · Reviewer_5qNA · 2024-11-01

**Soundness:** 2
**Presentation:** 2
**Contribution:** 2
**Rating:** 3
**Confidence:** 4

**Summary:**

This paper introduces a new method Multi-Agent Speedy Interpretable Machine Unlearning
(MASIMU) for efficiently “unlearning” data from AI models, essential for privacy compliance
without requiring complete model retraining. This approach combines deep learning,
reinforcement learning, and explainable AI(LIME) to remove specific data influences. It
fine-tunes a model on the retain set while re-weighting gradients to diminish the impact of data
needing removal (the forget set). By using multiple agents with reinforcement learning, MASIMU
breaks down complex image data into smaller, manageable parts, allowing faster and more
interpretable unlearning. This method has shown good performance in speed, robustness, and
interpretability across various datasets, including CIFAR-10, MNIST, and sensitive medical
images.

**Strengths:**

1. The authors have formulated a critical, state-of-the-art problem in AI by proposing a
method focused on enhancing trustworthiness and safeguarding the privacy of data.
2. The proposed method described in this paper is clear and easy to follow for
reproduction.
3. The MASIMU framework incorporate a multi-agent system to divide the unlearning tasks
among agents, which allows for faster processing and more manageable handling of
high-dimensional data

**Weaknesses:**

1. Since machine unlearning is not entirely a new concept, the author should compare the
results with at least one or more existing works that are similar, such as "SALUN:
Empowering Machine Unlearning via Gradient-Based Weight Saliency in Both Image
Classification and Generation," published in ICLR 2024.
2. The results presented in the paper do not sufficiently support the authors' claims. To
strengthen their findings, they should consider using well-known datasets like ImageNet
to demonstrate the effectiveness of their method. Additionally, including qualitative
examples of the model's behavior regarding the forgotten image set would enhance
clarity and impact.
3. A well-structured and state-of-the-art literature review on existing works in machine
unlearning is missing from the paper.
4. What is the motivation for performing machine unlearning on the selected datasets? i) You can easily train a new model within a few minutes or hours. ii) None of the datasets have an intuitive application to machine unlearning. Why not report results on the same dataset as the 2023 contest that the authors cite?

**Questions:**

There was a contest on this topic in 2023 at NeurIPS. Why have you not compared against those solutions?

---

### Official Review · Reviewer_JZEK · 2024-11-03

**Soundness:** 1
**Presentation:** 1
**Contribution:** 1
**Rating:** 1
**Confidence:** 3

**Summary:**

The paper introduces MASIMU, a framework that combines machine unlearning (MU) with a multi-agent reinforcement learning (MARL) setup for improved efficiency and interpretability. The method relies on two components. First, it leverages an interpretable AI (XAI) method to weight gradients based on the influence of data marked for removal (forget set) and their similarities to the retain set. Second, the authors introduce a multi-agent reinforcement learning (MARL) framework, in order to effectively manage high-dimensional data by having multiple agents communicate positional information and navigate image environments; this setup reduces each agent's observation space, allowing them to focus on unlearning critical gradients associated with key super-pixels that impact target labels. The authors performs experiments on MNIST, CIFAR-10, RESISC-45 (satellite images), and HAM-10000 (skin cancer images) and evaluate MASIMU’s performance in terms of robustness, interpretability, and speed.

**Strengths:**

The core ideas introduced by the authors are original and interesting:
- Utilizing a XAI method (LIME) to compare the similarities of the retain and forget samples and weight gradients accordingly is compelling, and might be able to improve unlearning benchmarks, as well as our understanding of it.
- The MARL framework proposed to reduce the input dimension and focus agents on unlearning interpretable gradients of important superpixels is also an interesting approach.
- The authors use various metrics in their evaluations (time, completeness, MIA) that go beyond the standard ones used.

**Weaknesses:**

My main concerns about the paper are the following:
- Presentation: it's very challenging to fully understand the methods used and all its components.
- Experiments: it seems that the comparisons are only between a baseline and variations of the framework introduced. I'd like to see how MASIMU performs against other state-of-the-art unlearning techniques.
- Complexity of the framework: the MARL approach introduced has a lot of complexity and computational overheads, raising questions about the generalizability of the approach on other settings.
- Vision-only: could the framework be extended to handle non-vision (for example language) tasks? If so, how?

**Questions:**

My questions are based on the weaknesses mentioned and how they could be improved:
- Is it possible to improve the presentation? E.g. give the high level ideas about the framework choices (why LIME?, why MARL?, what are the motivations?), figures and high-level descriptions about the approach, and clarifying various parts of the text and the algorithms. Ideally, the reader should be able to follow along the paper easily, and the techniques introduced should "make sense" to them.
- Can we add experiments of MASIMU against other state-of-the-art (SOTA) unlearning techniques?
- Can the authors address the question about the generalizability of their approach? How hard is it to make it work on a different setup, against the SOTA? How carefully should the hyper-parameters be chosen, and how big is the effort to do so?

---

### Official Review · Reviewer_FVPS · 2024-11-03

**Soundness:** 2
**Presentation:** 1
**Contribution:** 2
**Rating:** 3
**Confidence:** 4

**Summary:**

The paper introduces MASIMU, a framework aiming to enhance the efficiency, robustness, and interpretability of machine unlearning processes in deep learning models. The authors propose a method that purportedly outperforms existing unlearning approaches by integrating deep learning, multi-agent reinforcement learning (MARL), and explainable AI (XAI) techniques. The framework is evaluated on four datasets: CIFAR-10, MNIST, RESISC-45, and HAM-10000 to demonstrate its effectiveness in handling high-dimensional data and sensitive information.

**Strengths:**

1. The paper addresses the timely and significant issue of machine unlearning in the context of evolving privacy regulations and AI trustworthiness, which is crucial for applications involving sensitive data.
2. The attempt to combine deep learning, MARL, and XAI reflects an interdisciplinary approach, showcasing an effort to tackle the unlearning problem from multiple angles.
3. The use of various datasets, including high-resolution and medical images, indicates an effort to validate the method across different domains and data types.

**Weaknesses:**

1. It is unclear whether the authors conducted a comprehensive literature review of current state-of-the-art machine unlearning techniques [1-3]. Specifically, how does MASIMU differentiate itself from or improve upon existing methods? Additionally, the manuscript fails to provide comparisons between the MASIMU framework and other state-of-the-art machine unlearning techniques. Without such comparative analyses, it remains ambiguous whether MASIMU offers any substantive improvements. Why have the authors not conducted quantitative comparisons with other advanced machine unlearning methods? How does MASIMU perform in terms of efficiency, effectiveness, and scalability relative to these existing methods?

2. It is a recognized standard in machine unlearning research to use retraining as the benchmark for unlearning performance [1-5]. However, the paper does not provide such baseline results for reference.

3. The inherent trade-off between the ability to retain knowledge and the level of residual information pertaining to the forgotten data is not addressed. An in-depth discussion on this relationship would enhance the understanding of the proposed method’s effectiveness.

4. While the paper highlights the incorporation of LIME to achieve interpretability within the MASIMU framework, it lacks specific statistical explanations detailing this interpretability within the unlearning context. How does LIME's integration specifically enhance the interpretability of the unlearning process in MASIMU? The inclusion of visualizations or case studies demonstrating this interpretability and its benefits to the unlearning process would be valuable.

5. The paper does not provide a theoretical basis or analysis to underpin the proposed method. Critical components, such as the integration of MARL and XAI in the context of machine unlearning, are not theoretically justified. What is the theoretical foundation of the MASIMU framework? Can the authors provide theoretical proofs or analyses to substantiate claims regarding the improved efficiency, robustness, and interpretability of their approach?

6. The manuscript suffers from writing issues, including grammatical errors and unclear explanations, which hinder the reader's comprehension of the proposed method and its contributions. The figures are difficult to read (Fig. 3 and 4), and there are duplicate citations (e.g. 570-572). Have the authors conducted a thorough revision to improve the clarity and coherence of the paper? Additionally, could the authors consider reorganizing the structure to present the methods and results in a more logical and clear manner?

7. The analysis of experimental results is insufficient, as it lacks comparisons with relevant baselines and detailed discussions on the evaluation metrics used. Important details such as hyperparameter settings and statistical significance are omitted. Furthermore, the authors have not provided the source code, which would facilitate implementation and further validation of their work.

> [1] Fan, Chongyu, et al. "SalUn: Empowering Machine Unlearning via Gradient-based Weight Saliency in Both Image Classification and Generation." The Twelfth International Conference on Learning Representations.
>
> [2] Liu, Jiancheng, et al. "Model sparsity can simplify machine unlearning." Advances in Neural Information Processing Systems 36 (2024).
>
> [3] Kurmanji, Meghdad, et al. "Towards unbounded machine unlearning." Advances in neural information processing systems 36 (2024).
>
> [4] Golatkar, Aditya, Alessandro Achille, and Stefano Soatto. "Eternal sunshine of the spotless net: Selective forgetting in deep networks." Proceedings of the IEEE/CVF Conference on Computer Vision and Pattern Recognition. 2020.
>
> [5] Thudi, Anvith, et al. "Unrolling sgd: Understanding factors influencing machine unlearning." 2022 IEEE 7th European Symposium on Security and Privacy (EuroS&P). IEEE, 2022.

**Questions:**

See weakness.

---

### Official Review · Reviewer_Utve · 2024-11-04

**Soundness:** 2
**Presentation:** 2
**Contribution:** 2
**Rating:** 3
**Confidence:** 3

**Summary:**

This paper introduces Multi-Agent Machine Unlearning (MAMU), a multi-agent framework for machine unlearning, which aims to remove specific data points from trained models without complete retraining. The framework uses multiple agents that work collaboratively using recurrent neural networks (RNNs) to traverse and process images. The authors present four variants of their framework: MALMU (LSTM-based), MASMU (GRU-based), and their interpretable counterparts MALIMU and MASIMU. The approach is evaluated on high-dimensional data like images, examining both unlearning speed and model accuracy on retained data.

**Strengths:**

- Unlearning is an important field, which will be necessitated as privacy regulations and "right to be forgotten" requirements become more prevalent in real-world ML applications
- The approach of using MARL for unlearning is (to my knowledge at least) novel
- Evaluations cover a wide variety of datasets, including medical

**Weaknesses:**

- There is no dedicated related work section, and the introduction itself is quite bare. I recommend adding an explicit related work section
- There is no comparison to existing state-of-the-art unlearning methods making it impossible to evaluate the claimed advantages of MASIMU in the context of current unlearning literature
- Results do not include statistical significance tests, and performance metrics are quite close in many cases, leading to uncertainty about whether the proposed method offers meaningful improvements over baselines

**Questions:**

- **Line 254**: Apart from the unlearning reweighing, how is your method different from Mousavi et al., 2019a? The multi agent algorithm you are using seems to be theirs, so I would not claim novelty
- **Line 379**: Which results in this table are statistically significant? Also the presentation of this table could be improved e.g. is lower or higher better for MIA / COMP
- **Line 401**: It's not obvious the value of plotting the accuracy / loss curves in the main paper. This would be better left for the appendix and would provide room for a proper related works section
- **Line 433**: Similarly to table 1, presentation in this table is poor. On its merits, it lacks statistical significance, and the results look very close to each other. Which results are statistically significant?
- **Line 461**: Similarly here the unlearning accuracy and loss plots do not provide much value. What are you trying to show by providing these plots in the main paper?

---

### Note · Authors · 2024-11-20

I have read and agree with the venue's withdrawal policy on behalf of myself and my co-authors.